# eNOS-NO-induced small blood vessel relaxation requires EHD2-dependent caveolae stabilization

Claudia Matthaeus[1]*, Xiaoming Lian[2], Séverine Kunz[3], Martin Lehmann[4], Cheng Zhong[2], Carola Bernert[1], Ines Lahmann[5], Dominik N. Müller[6], Maik Gollasch[2,7], Oliver Daumke[1,8]*

**1** Crystallography, Max-Delbrück-Center for Molecular Medicine, Berlin, Germany, **2** Charité—Universitätsmedizin Berlin, Experimental and Clinical Research Center (ECRC), Campus Buch, Berlin, Germany, **3** Electron Microscopy Core Facility, Max-Delbrück-Center for Molecular Medicine, Berlin, Germany, **4** Department of Molecular Pharmacology & Cell Biology and Imaging Core Facility, Leibniz-Forschungsinstitut für Molekulare Pharmakologie, Berlin, Germany, **5** Signal Transduction/Developmental Biology, Max-Delbrück-Center for Molecular Medicine, Berlin, Germany, **6** Experimental & Clinical Research Center, a cooperation between Charité Universitätsmedizin Berlin and Max Delbrück Center for Molecular Medicine, Berlin, Germany, **7** Charité—Universitätsmedizin Berlin, Medical Clinic for Nephrology and Internal Intensive Care, Campus Virchow, Berlin, Germany, **8** Institute of Chemistry and Biochemistry, Freie Universität Berlin, Berlin, Germany

* claudia.matthaeus@mdc-berlin.de (CM); oliver.daumke@mdc-berlin.de (OD)

**Data Availability Statement:** All relevant data are within the manuscript, Supporting Information files, and on OSF: https://osf.io/rbvjz/. DOI: 10.17605/OSF.IO/RBVJZ.

## Abstract

Endothelial nitric oxide synthase (eNOS)-related vessel relaxation is a highly coordinated process that regulates blood flow and pressure and is dependent on caveolae. Here, we investigated the role of caveolar plasma membrane stabilization by the dynamin-related ATPase EHD2 on eNOS-nitric oxide (NO)-dependent vessel relaxation. Loss of EHD2 in small arteries led to increased numbers of caveolae that were detached from the plasma membrane. Concomitantly, impaired relaxation of mesenteric arteries and reduced running wheel activity were observed in EHD2 knockout mice. EHD2 deletion or knockdown led to decreased production of nitric oxide (NO) although eNOS expression levels were not changed. Super-resolution imaging revealed that eNOS was redistributed from the plasma membrane to internalized detached caveolae in EHD2-lacking tissue or cells. Following an ATP stimulus, reduced cytosolic $Ca^{2+}$ peaks were recorded in human umbilical vein endothelial cells (HUVECs) lacking EHD2. Our data suggest that EHD2-controlled caveolar dynamics orchestrates the activity and regulation of eNOS/NO and $Ca^{2+}$ channel localization at the plasma membrane.

## Introduction

Caveolae are small plasma membrane invaginations of 80 nm in size. They are most abundantly found in adipocytes and endothelial cells in which they compensate for mechanical membrane tension [1–4]. Caveolae are also involved in endocytic processes like cellular lipid uptake [5–8] or cholera toxin internalization [9–11]. In endothelial cells, Caveolin1 (Cav1) negatively regulates the activity of endothelial nitric oxide synthase (eNOS) [12–14]. In

**Funding:** This work was supported by- to OD: Sonderforschungsbereich SFB958, project A12, http://www.sfb958.de/, Helmholtz-Gemeinschaft, Initiative and Networking Fund. To MG: Deutsche Forschungsgemeinschaft DFG research grant 318527103, http://gepris.dfg.de/gepris/person/1421037?context=person&task=showDetail&id=1421037&. The funders had no role in study design, data collection and analysis, decision to publish, or preparation of the manuscript.

**Competing interests:** The authors have declared that no competing interests exist.

addition, caveolae accommodate a variety of proteins involved in calcium influx and signaling (reviewed in ref. [15,16]). For example, the transient receptor potential cation channels (TRPC) TRPC1 and 4 channels localize to caveolae and facilitate $Ca^{2+}$ influx within the store-operated $Ca^{2+}$ entry (SOCE) channel complex [17–20].

Caveolins (Cav1-3) and cavin proteins (cavin1-4) are the major caveolar coat proteins and are essential for the generation of caveolae (reviewed in ref. [2,3,21–23]). In addition, the BAR-domain containing protein PACSIN2/syndapin2 [24–26] or PACSIN3 [27] generate membrane curvature during caveolar biogenesis and may stabilize the caveolar neck. A fourth structural component of caveolae is the Eps15 homology domain containing protein 2 (EHD2) [9,28–30]. EHD2 belongs to the dynamin-related EHD family [31,32] that comprises four members in human and mice. We and others previously showed that loss of EHD2 results in increased caveolar turnover rate and mobility and detachment of caveolae from the plasma membrane [9,29,33]. In adipocytes and fibroblasts, loss of EHD2 leads to increased fatty acid uptake, suggesting that EHD2 may be a negative regulator of caveolae function [33]. Based on the structural and cellular data, we suggested that EHD2 forms ring-like oligomers surrounding the neck of caveolae, therefore stabilizing caveolae at the plasma membrane [34].

Endothelial nitric oxide synthase (eNOS) is abundantly expressed in endothelial cells that form the inner layer of blood and lymphatic vessels [35]. An eNOS homodimer comprised of two 138 kD subunits catalyzes the production of L-citrulline and nitric oxide (NO) from L-arginine and molecular oxygen [35,36]. eNOS-derived NO diffuses across the plasma membrane and activates soluble guanylate cyclase in the surrounding smooth muscle cells leading to cGMP production and consequently to vessel relaxation. In addition, NO activates angiogenesis and smooth muscle proliferation. When released into the blood vessel lumen, NO inhibits platelet aggregation on the vessel wall [35].

The involvement of eNOS and NO in a variety of endothelial signaling pathways implies that its activity and localization must be tightly regulated. eNOS localizes to cholesterol-rich domains of the plasma membrane [37,38] and its activity is inhibited by binding to Cav1 [13,39]. In turn, various external stimuli, including vascular endothelial growth factor (VEGF), acetylcholine or insulin lead to activation of eNOS by mediating its release from Cav1/caveolae into the cytosol [40,41]. For example, eNOS can be activated by phosphorylation of Ser-1177 via AKT/protein kinase B, protein kinase A (PKA), $Ca^{2+}$/calmodulin-dependent protein kinase II (CamKII) [42–44] or AMP-activated protein kinase (AMPK). Alternatively, dephosphorylation of the inhibitory sites (e.g. Thr-495 or Ser-113), for example by protein phosphatase 2A, also leads to eNOS activation [41]. Also $Ca^{2+}$-loaded calmodulin disrupts the Cav1-eNOS interaction by directly binding to eNOS, leading to eNOS activation [40,44].

Here, by characterizing an EHD2 knockout mouse model, we describe a novel function of EHD2 and caveolae related to eNOS regulation. In endothelium, loss of EHD2 results in the detachment of caveolae from the plasma membrane and inhibition of eNOS. Mice lacking EHD2 show reduced vessel relaxation due to impaired NO production. Furthermore, in human umbilical vein endothelial cells (HUVECs), we observe reduced cytosolic $Ca^{2+}$ influx after EHD2 knockdown. Our data indicate that EHD2-dependent caveolae plasma membrane stabilization regulates correct eNOS localization and function as well as stimulated $Ca^{2+}$ responses and is therefore involved in the physiological function of blood vessels.

## Methods & materials

### EHD2 delta E3 mouse strain

A mouse strain lacking exon 3 of EHD2 was generated as described previously [33]. After backcrossing with C57BL/6N for 7 generations, EHD2 del/+ and EHD2 del/del mice were

breed and newborn mice were genotyped. EHD2 del/+ mice were used as control group to reduce animal numbers. All mice experiments were performed with adult male mice (>20 weeks) accordingly to German Animal protection law (animal application Berlin LaGeSo G0154/14).

## Primary mesenteric artery culture

EHD2 del/+ and del/del mice were sacrificed by cervical dislocation, and mesenteric arteries were carefully dissected. After removal of surrounding white adipose tissue (WAT), the arteries were cut in small pieces and cultivated on Matrigel (Sigma) in endothelial growth medium/ 10%FBS and endothelial supplement (Promocell). Every second day, the medium was changed and newly formed vessels were imaged by Zeiss Axiovert100.

## HUVEC cell culture

HUVEC (passage 2) were a kind gift from Holger Gerhard from Max-Delbrück-Center. The cells were cultivated in Endothelial growth medium (EM, Promocell) added with endothelial supplement (Promocell) and 10% FBS and 1% streptomycin/penicillin (Gibco) at 37°C and 5% $CO_2$. HUVEC were split with 0.01% trypsin/EDTA/PBS (Gibco) every second day and seeded in fresh EM.

## HUVEC siRNA knockdown

Freshly split HUVECs (passage 2–6) were electroporated for EHD2 siRNA knockdown with the GenePulser XCell (Biorad). Briefly, HUVECs were split as described before and the obtained cell pellet was resuspended in OptiMEM/10%FBS (Gibco). After cell counting, the HUVEC cell suspension was diluted to $1.8x10^6$ cells/ml and 300 μl were transferred into electroporation cuvettes (2 mm, Biorad). EHD2 stealth siRNA (Thermo Fisher) and siRNA negative control (medium GC content) were added to a final concentration of 300 nM. After mixing, the cuvettes were placed into the electroporation device and the pulse (160 μOHM, 500 μF, ∞ resistance) was applied. The electroporated cells were cultivated in EM/10%FBS for 48 h and EHD2 siRNA knockdown was analyzed by EHD2 antibody staining and Western blot.

## Echocardiography

In vivo heart function was analyzed by high frequency ultrasound imaging in B and M-mode (non-invasive, 2D conventional echocardiography) of 10, 20 and 52 week old EHD2 del/+ and del/del mice. Mice were anaesthetized with isoflurane, the fur was shaved and ultrasound imaging was performed with Vevo2100 transducer system (VisualSonics).

## Blood pressure measurement

The blood pressure was measured by the non-invasive tail-cuff method as previously described. Briefly, EHD2 del/+ and del/del mice were placed in a mouse restrainer of the CODA system (Kent Scientific) to prevent extensive movement. A cuff was applied around the tail and blood pressure was recorded by the CODA system.

## Running wheel activity

EHD2 del/+ and del/del mice were singly placed in cages containing a voluntary running wheel for 2 weeks, in which the running distance (in km) and running activity were monitored. All inspected mice did not run during the day. Therefore, the recorded running activity

was analyzed exclusively during the night/ dark phase. The total distance after 1, 4, 8, 11 and 14 days (and nights) was monitored by using a bicycle computer attached to the running wheel. The activity was tracked by monitoring each rotation of the wheel and is displayed as count per logging interval (measurement every 5 min).

## Wire myography

The second branches of mesenteric arteries were isolated from EHD2 del/+ and del/del mice under inhalation anesthesia with isoflurane by cervical dislocation and then quickly transferred to cold (4°C), oxygenated (95% $O_2$/5% $CO_2$) physiological salt solution (PSS) containing 119 mM NaCl, 4.7 mM KCl, 1.2 mM $KH_2PO_4$, 25 mM $NaHCO_3$, 1.2 mM $MgSO_4$, 11.1 mM glucose, 1.6 mM $CaCl_2$. After cleaning the connective tissue with scissors without damaging the adventitia, the arteries were dissected into 2 mm rings. Each ring was positioned between two stainless steel wires (diameter 0.0394 mm) in a 5 mL organ bath of a Small Vessel Myograph (DMT 610M, Danish Myo Technology, Aarhus, Denmark). The organ bath was filled with PSS. The bath solution was continuously oxygenated (95% $O_2$/5% $CO_2$), and kept at 37°C (pH 7.4). COX inhibitors were not added as prostaglandin generation is strongly reduced in small mesenteric arteries [45,46]. The rings were placed under a tension equivalent to that generated at 0.9 times the diameter of the vessel at 100 mmHg (DMT Normalization module by CHART software). The software Chart5 (AD Instruments Ltd. Spechbach, Germany) was used for data acquisition and display. The rings were pre-contracted and equilibrated for 30 min until a stable resting tension was acquired. Thereafter, the rings were constricted with 1 μM phenylephrine (PE) to ensure viability and allow normalization of the relaxant response by acetylcholine (ACh, 1 μM) to confirm the eNOS functionality. In the end, the rings were constricted with isotonic external 60 mM potassium chloride (KCl) to again ensure the viability of arterial rings. The NO donor SNP was applied in different concentrations after PE (1 μM) application and relexation was recorded. The composition of isotonic external 60 mM KCl was 63.7 mM NaCl, 60 mM KCl, 1.2 mM $KH_2PO_4$, 25 mM $NaHCO_3$, 1.2 mM $Mg_2SO_4$, 11.1 mM glucose, and 1.6 mM $CaCl_2$.

## Histology

EHD2 del/+ and EHD2 del/del mice were anesthetized with 2% ketamine/10%rompun, perfused by 30 ml PBS and 50 ml 4% PFA and afterwards vessels or hearts were dissected. After 24 h of fixation in 4% PFA, tissues were dehydrated in 70–100% EtOH and incubated in xylol (Merck) for 48 h. Next, the tissues were embedding in paraffin, 4 μm paraffin sections were obtained, and Masson Trichrome staining (Kit, Sigma) was applied accordingly to manufacturer´s protocol. Images were obtained at Zeiss Axiovert100 microscope.

## Immunohistostaining of cryostat sections for confocal and STED imaging

Perfused and fixed EHD2 del/+ and EHD2 del/del mice (as described before) were dissected and the investigated tissue pieces were further fixed for 4 h in 4% PFA, transferred to 15% sucrose (in PBS, Merck) for 4 h and finally incubated overnight in 30% sucrose. After embedding in TissueTek, the tissue is frozen at -80°C and 10 μm sections were obtained in a Leica cryostat at -30°C. For immunostainings, the cryostat sections were incubated with blocking buffer (1% donkey serum/1% TritonX100/PBS, STED: 1% donkey serum/0.1% TritonX100/PBS) before the first antibody was applied overnight at 4°C. After washing with PBS/1% Tween, the secondary antibody and DAPI was applied for 2 h, and afterwards the sections were embedded in ImmoMount (ThermoScientific #9990402) or STED imaging buffer (Pro-Long Gold antifade reagent, Molecular Probes #P36934). The stained sections were analyzed

with Zeiss LSM700 microscope provided with Zeiss objectives 5, 10, 20, 40 and 63x, Zeiss LSM880 Airy Scan with 63x airy objective or Leica SP8 STED microscope with 100x objective. The obtained images were further investigated by ZEN or Leica software and ImageJ/Fij. STED images displayed in Fig 3E and 3F were deconvoluted by Huygens Software. eNOS staining intensity (Integrated Intensity value) of EHD2 del/+ and del/del mesenteric arteries was determined at different plasma membrane regions (300–400 μm$^2$) of the investigated sections by ImageJ. 3 different mice/genotype were used to determine the mean eNOS staining intensity value, as depicted in Fig 3E. Antibodies: anti-Cav1-rabbit (abcam #2910, 1:200), anti-Cav1-mouse (Sigma #SAB4200216, 1:200), anti-eNOS-mouse (BD Transduction Laboratories #610297, 1:200–1:500), anti-EHD2-rabbit (self-made, previously used in [33]), anti-Muscarinic Acetylcholine receptor M3-rabbit (abcam #ab150480, 1:80), anti-mouse-Alexa488 (Invitrogen #R37114, 1:500), anti-rabbit-Cy3 (Dianova #711-165-152, 1:500), DAPI (Sigma, 1:1000), anti-rabbit-Atto647 (Invitrogen #A27040, 1:800), anti-mouse-Atto546 (Invitrogen #A11030, 1:800).

### DAF staining

Determination of NO generated in mesenteric arteries or HUVEC was done by DAF staining as previously described [47,48]. Briefly, freshly prepared mesenteric arteries were cut in small pieces and incubated for 30 min in physiological buffer (125 mM NaCl, 5 mM KCl, 1 mM MgSO$_4$, 1 mM KH$_2$PO$_4$, 6 mM glucose, 25 mM HEPES, 2 mM CaCl$_2$, pH 7.4, all obtained from Merck) at 37˚C. 10 μM DAF-2FM (Molecular probes #D23842, diluted in DMSO) was added and arteries were incubated for further 30 min at 37˚C, followed by 100 μM acetylcholine treatment and 10 min incubation. Afterwards, the tissue samples were embedded in TissueTek and flash frozen. To investigate DAF staining, frozen tissues were cryosectioned and counterstained with DAPI. HUVEC cells were washed with physiological buffer and 10 μM DAF-2FM was added for 30 min. After 10 min of 100 μM acetylcholine treatment, the cells were stained with DAPI, fixed with 4% PFA for 10 min and embedded in ImmoMount. Mesenteric arteries or HUVEC were incubated with 100 μM L-NAME for 30 min before DAF-2FM was added to the buffer. The staining was analyzed with confocal imaging Zeiss LSM700 and 40x or 63x objective.

### Calcium imaging

HUVEC cells were seeded on fibronectin coated glass dishes and after 2 days in culture incubated with Fura2-AM (Biomol, diluted in DMSO, 1 μg/ml) for 60 min at 37˚C in culture medium. Glass dishes were transferred to the imaging chamber including pre-warmed Krebs buffer (125 mM NaCl, 5 mM KCl, 1 mM MgSO$_4$, 1 mM KH$_2$PO$_4$, 6 mM glucose, 25 mM HEPES, 2 mM CaCl$_2$, pH 7.4, all obtained from Merck). Fluorescence intensities (FI) recordings were performed with Zeiss Axio Microscope, 63x water objective and the following settings: alternating excitation at 340 nm and 380 nm, 200 ms exposure time, cycle time 2 s, total recording 5 min, 3x3 binning. Cytosolic Ca$^{2+}$ is calculated from FI(340 nm)/FI(380 nm). 30 μM ATP (Biomol) was freshly prepared in Krebs buffer and applied by perfusion into the imaging chamber during recordings. Data was analyzed by ImageJ/Fiji and GraphPad Prism.

### Transmission Electron microscopy (TEM)

Mice were fixed by perfusion with 4% (w/v) formaldehyde in 0.1 M phosphate buffer and hearts were dissected to 1–2 mm$^3$ cubes. Tissue blocks and HUVEC cells were fixed in phosphate buffered 2.5% (v/v) glutaraldehyde for 1–2 hours. Cells were pelleted, stabilized by 10% (w/v) gelatin and further processed as 1–2 mm$^3$ cubes. Additional membrane contrast was

added first by 1% tannic acid in 0.1 M cacodylate buffer and second by aqueous 2% uranylacetate. After 1% (v/v) osmium tetroxide treatment, tissue and cells cubes were dehydrated in a graded series of ethanol and embedded in the PolyBed® 812 resin (Polysciences Europe GmbH), ultrathin sections (60–80 nm) were cut (Leica microsystems) and uranyl acetate and lead citrate staining was performed. Samples were examined at 80 kV with a Zeiss EM 910 electron microscope (Zeiss), and image acquisition was performed with a Quemesa CDD camera and the iTEM software (Emsis GmbH).

### *In situ* hybridization

In situ hybridizations were performed on 14 μm cryosections prepared from E18.5 C57BL/6N embryos as previously described [49]. Digoxigenin-labeled riboprobes were generated using a DIG-RNA labeling kit (Roche), whereby EHD2 specific in situ probe was generated by PCR amplification of a 400 bp fragment from C57BL/6N cDNA (EDH2_ISH_FWD: 5'-CAGGT CCTGGAGAGCATCAGC-3'; EDH2_ISH_REV: 5'- GAGGTCCTGTTCCTCCAGCTCG-3'). The PCR product was cloned into pGEM-Teasy (Promega). T7 and sp6 polymerases were used to generate Ehd2-sense and antisense probes.

### Western blot

EHD2 del/+ and EHD2 del/del mice were sacrificed by cervical dislocation and mesenteric arteries were dissected and tissue disruption was performed in 1x RIPA buffer (Abcam) with a glass homogenizer, followed by 5 min ultrasonication and 10 min centrifugation at maximal speed. HUVECs were washed with ice-cold PBS and homogenized in 1xRIPA buffer, followed by 5 min ultrasonication and 10 min full speed centrifugation. Supernatant was transferred in a fresh tube and protein concentration was analyzed by SDS-PAGE followed by Commassie staining. At least 10 μg protein/lane was applied to 4–12% SDS-PAGE NuPage (Invitrogen) and SDS-PAGE was performed according to the manufacture's protocol with MOPS buffer (Invitrogen). Afterwards, proteins were blotted on nitrocellulose membrane (Amersham) at 80 V for 1 h, followed by blocking of the membranes with 5% milk powder (in TBST, 150 mM NaCl, 20 mM Tris-HCl, pH 7.5, 0.1% Tween20) for 2 h at room temperature. To detect specific protein level antibodies against protein of interest (1:500–1:1000) was applied over night at 4°C. After washing with TBST, the secondary antibody labelled with HRP was applied to the membrane for 2 h. Detection of protein bands resulted from ECL detection solution and images were obtained by ChemiDoc XRS (Biorad). The following antibodies were used: anti-eNOS-mouse (BD Transduction Laboratories #610297, 1:200), anti-phospho-Ser1177-eNOS-rabbit (Cell Signaling #9571, 1:100), anti-phospho-AKT Ser473 and Thr308 (both raised in rabbit, AKT kit, Cell Signaling #9916, 1:100), anti-EHD2-rabbit (self-made, previously used in [33]), anti-calmodulin-mouse (Invitrogen #MA3-917, 1:1000), anti-GAPDH-mouse (Novus Biologicals #NB300-221, 1:2000), anti-beta-actin-mouse (Sigma #A1978, 1:2000), anti-rabbit-HRP and anti-mouse-HRP (both Dianova, 1:10000).

### Statistical analysis

All applied statistical tests were calculated with Prism (GraphPad software). Normality distribution test (Kolmogorov-Smirnov test) was carried out for all experimental values, and if the data was normally distributed, Student t-Test (two-tailed P-value) was applied, otherwise Mann-Whitney-Rank-Sum (two-tailed P-value) test was used to calculate the significant difference between two groups. EHD2 and eNOS staining intensities were correlated for C57BL/6N mesenteric arteries (Fig 1B) by using Pearson correlation. Box plots, if not otherwise indicated in the figure legends, always represents median with whiskers from minimum to maximum,

column bar graphs and line graphs represent mean with mean standard error of the mean (SE). For all experiments including the examination of mice or mouse tissue, n represents the number of mice which were used (Figs 1, 2, 3 and 6, S1, S2 and S3 Figs) and all analyzed cryo/paraffin sections or caveolae are also indicated (e.g.: n = 80 caveolae/3 mice). For HUVEC cell culture experiments (Figs 4 and 5, S4 Fig) n represents the number of observed events (e.g.: staining intensity) and the number of independently performed experiments (e.g.: n = 80 measured DAF staining intensities/3 independent experiments). The following P-values were used to indicate significant difference between two groups: * $P<0.05$; ** $P<0.001$; *** $P<0.0001$.

## Results

### EHD2 expression in endothelial cells is essential for caveolae stabilization at the plasma membrane

EHD2 antibody staining in cryostat sections of adult C57BL/6N brown adipose tissue revealed high EHD2 expression in blood vessels (Fig 1A). Similarly, aorta cryostat sections showed a strong EHD2 signal co-localized with the endothelial marker protein eNOS (Fig 1B), in contrast to the complete lack of EHD2 staining in EHD2 del/del mesenteric arteries (S1 Fig). Also endothelial cells of lymphatic vessels displayed high expression levels of EHD2 (Fig 1C). In situ hybridization of EHD2 mRNA in E18.5 C57BL/6N embryo corroborated the antibody staining results (Fig 1D). In line with results from other cell types [9,28], Cav1 and EHD2 showed a striking co-localization in cryostat sections of small mesenteric arteries (Fig 1E) and HUVECs (Fig 1F).

We previously described that adipocytes and fibroblasts in the EHD2 knockout mouse model contain an increased number of caveolae that are detached from the plasma membrane [33]. In line with the results obtained from adipocytes, endothelial caveolae were more often detached from the plasma membrane in EHD2 del/del compared to EHD2 del/+ mice (control group), as analyzed by electron microscopy (EM, Fig 1G and 1H). The absolute number of caveolae in small vessels lacking EHD2 was only slightly increased.

### Loss of EHD2 results in impaired vessel relaxation

Our localization experiment and ultrastructural analysis suggested a function of EHD2 in blood vessels. As previously reported [33], loss of EHD2 did not impair embryonic development, further illustrated by the Mendelian distribution of EHD2 knockout offspring (S2A Fig). Furthermore, we cultivated small mesenteric artery pieces obtained from EHD2 del/+ and del/del mice and observed formation of new vessels in both genotypes (S2B Fig). These results argue against a major role of EHD2 in angiogenesis.

EHD2 del/+ and del/del mesenteric arteries were dissected and histological analyses revealed no gross changes in morphology (Fig 2A) and diameter (Fig 2B). Active force measurements of isolated small mesenteric artery rings revealed a similar contraction of EHD2 del/del and del/+ arteries induced by phenylephrine (PE) and KCl (Fig 2C and 2E, see S3A Fig for complete KCl recovery traces). Strikingly, EHD2 del/del arteries showed a complete loss of acetylcholine (ACh)-induced relaxation in compared to immediate responses of acetylcholine on EHD2 del/+ mesenteric arteries (Fig 2C and 2D). Notably, muscarinic acetylcholine receptor M3 protein level assessed by immunohistostaining was not altered EHD2 del/del mesenteric arteries compared to EHD2 del/+ (S3B Fig).

As relaxation of blood vessels is related to eNOS and NO generation [50], the mesenteric artery rings were pre-treated with the eNOS inhibitor L-NAME. Under these conditions, EHD2 del/+ mesenteric arteries revealed impaired relaxation ($4.9 \pm 0.9\%$ vs. untreated

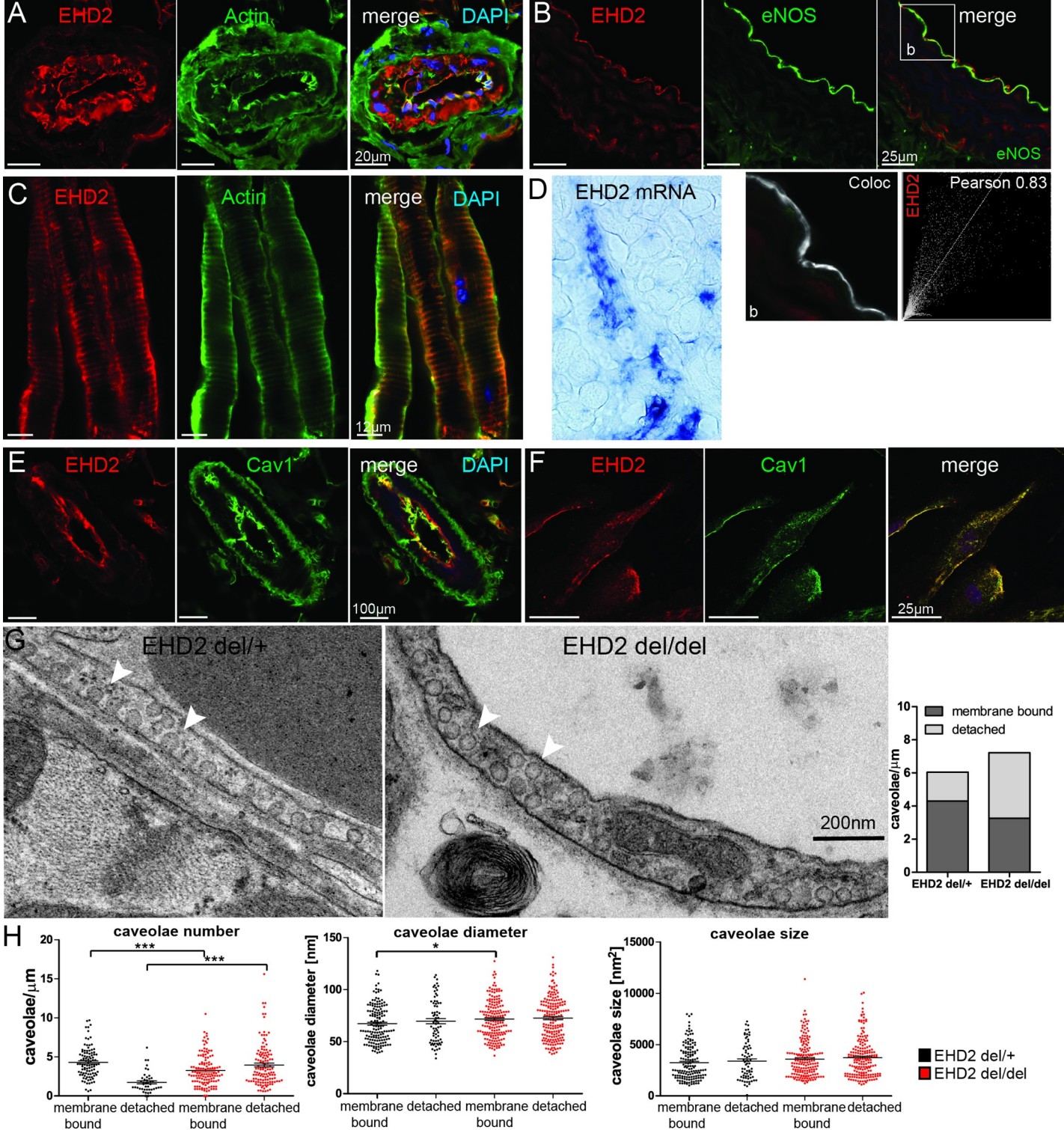

**Fig 1. EHD2 is expressed in endothelial cells and stabilizes caveolae at the plasma membrane.** (A) Cryostat section of adult C57BL/6N small arteries in brown adipose tissue stained against EHD2 and actin. (B) EHD2 and eNOS stained in aorta cryostat sections obtained from C57BL/6N mice illustrating co-localization of both proteins within the endothelium. (C) Cryostat sections of lymphatic vessels were stained against EHD2 and actin. (D) In situ hybridization of a C57BL/6N E18.5 embryo showed strong EHD2 expression in blood vessels. Cryostat sections of adult C57BL/6N mesenteric arteries (E) and HUVEC cells (F) stained against Cav1 and EHD2 revealed EHD2 localization at caveolae. (G) Representative EM images of EHD2 del/+ and EHD2 del/del arteries in which caveolae number, size and diameter were determined. (H) EHD2 del/del mice showed an increased number of detached caveolae compared to EHD2 del/+, in which mainly membrane-bound caveolae were

found (n(del/+) = 142/3; n(del/del) = 251/3, graphs illustrate each replicate with mean +/- SE, column bar graph illustrates mean, t-test or Mann Whitney U test were used to calculate significance, * P<0.05; ** P<0.001; *** P<0.0001). White arrows point to caveolae.

58 ± 3% relaxation, relative to PE-induced contraction monitored after 3 min) due to the inhibition of NO production (Fig 2F and 2H). In contrast, L-NAME did not affect the contraction loss in EHD2 del/del arteries (Fig 2G and 2H). This suggests that the reduced relaxation capability in arteries lacking EHD2 is caused by an impaired eNOS/NO system.

To analyze if the eNOS/NO downstream signaling pathways (e.g. cGMP/protein kinase G pathways) are still functional in EHD2 lacking arteries, the NO donor sodium nitroprusside (SNP) was applied to EHD2 del/+ and del/del mesenteric arteries. In these experiments, EHD2 del/+ and del/del mesenteric arteries did not show any difference in their relaxation ability upon SNP treatment (Fig 2I) indicating that the reduced relaxation of EHD2 del/del arteries in response to ACh results from reduced eNOS enzyme activity but not from impaired eNOS downstream signaling. Of note, no difference in resting blood pressure was observed in EHD2 del/+ and del/del mice by using tail-cuff blood pressure measurements (Fig 2J and 2K).

Subsequently, we investigated the function of the eNOS/NO system in mesenteric arteries in more detail. By staining with the fluorescent dye DAF [47,48], NO can be directly visualized in small arteries. Indeed, cryosections of EHD2 del/+ mesenteric arteries showed strong NO-DAF staining in the endothelium. In contrast, we observed a reduced DAF staining intensity in small vessels dissected from EHD2 del/del mice compared to those from EHD2 del/+ mice, pointing to a decreased NO concentration in the vascular endothelium lacking EHD2 (Fig 3A and 3B). Consistent with the previous active force measurements, pre-treatment of vessels with L-NAME strongly reduced DAF staining in EHD2 del/+ mice by 77%.

We reasoned that reduced NO concentration in EHD2 del/del mice may be caused by an impairment of eNOS activity or regulation. Therefore at first, eNOS localization in cryostat sections of mesenteric arteries was investigated by stimulated emission depletion microscopy (STED). In line with previous results, EHD2 del/+ arteries showed intense eNOS staining in the plasma membrane in close proximity to Cav1 (Fig 3C, see co-localization). In contrast, eNOS staining at the plasma membrane was reduced in EHD2 del/del arteries (Fig 3D and 3E), although the total protein level of artery tissue lysates was not changed (Fig 3F). Instead, eNOS was redistributed from the membrane to the cytosol in EHD2 del/del arteries (Fig 3D and S4 Fig).

To further dissect the cellular mechanism of impaired eNOS/NO signaling and to determine eNOS localization in the absence of EHD2 in more detail, we resorted to the analysis of human umbilical vein endothelial cells (HUVECs). EHD2 expression was knocked down in HUVECs by specific siRNA, whereas a non-sense siRNA served as a negative control. Reduced EHD2 staining intensity in immuno-fluorescence experiments (S5A and S5B Fig) and reduced EHD2 protein levels (S5C Fig) confirmed the successful knockdown approach. Subsequently, HUVECs were stained against Cav1 and eNOS and the localization of eNOS was investigated by STED microscopy (Fig 4A). In confocal images, no striking difference of eNOS staining intensity was observed in HUVECs treated with EHD2 siRNA compared to control cells (Fig 4A). Within the STED images, membrane-bound caveolae appeared as ring-like Cav1 structures, in contrast to detached caveolae that were observed as filled Cav1 vesicles (Fig 4B and 4C). Cav1 fluorescence intensity plot profiles and Cav1 surface plots further illustrated the characteristic caveolae shapes for membrane-bound and detached caveolae (Fig 4D). Compared to EHD2 knockdown HUVECs, more ring-like Cav1 structures were found in control cells (Fig 4A). Importantly, detached caveolae in EHD2 siRNA HUVECs showed distinct eNOS staining suggesting that eNOS still localizes to Cav1 in these caveolae (Fig 4C and 4D).

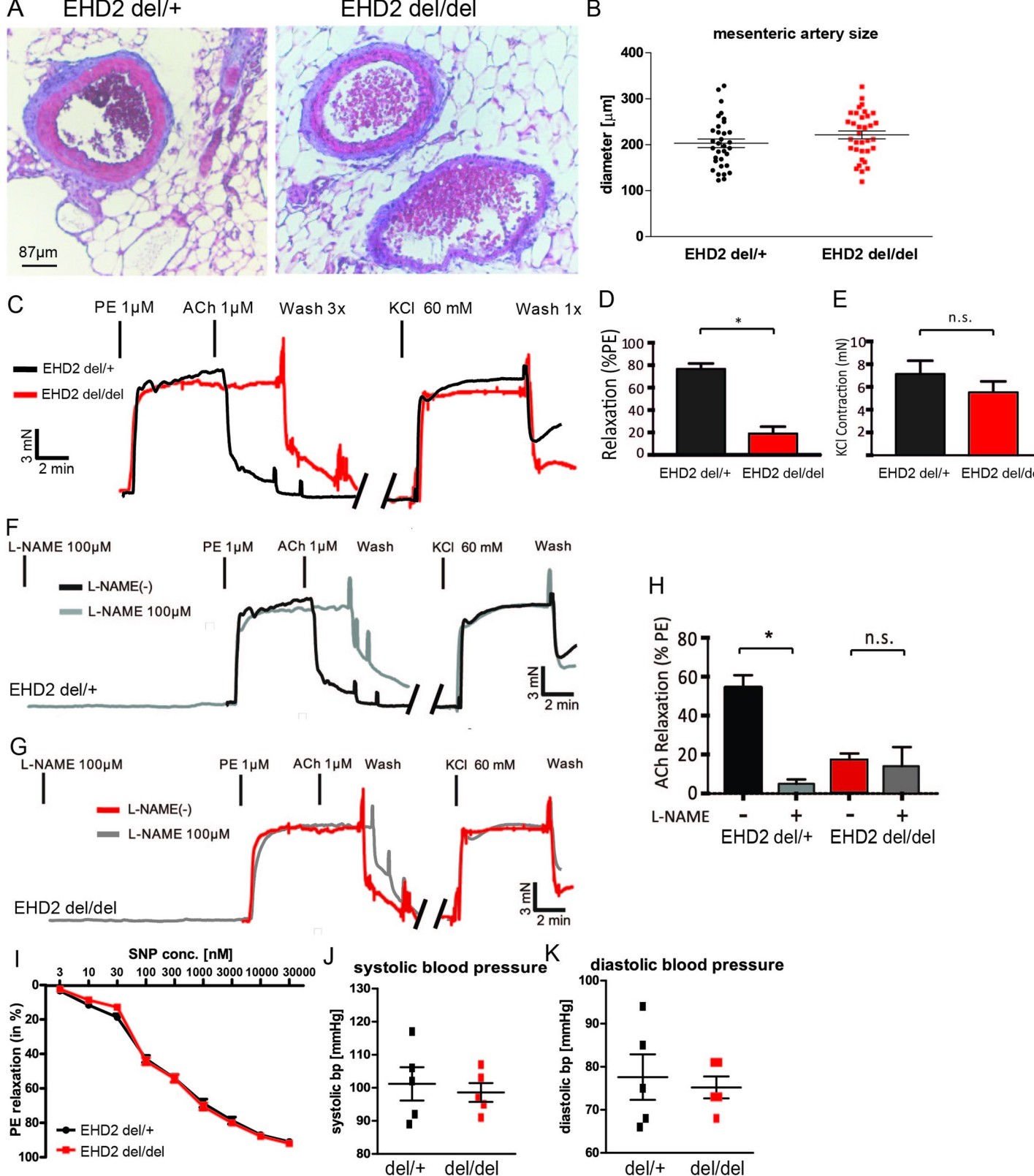

**Fig 2. Loss of EHD2 results in impaired relaxation of mesenteric arteries.** (A) Representative histological staining of EHD2 del/+ and del/del mesenteric arteries revealed no severe morphological differences. (B) Summary of mesenteric artery diameter in EHD2 del/+ and del/del mice (n(del/+) = 33; n(del/del) = 35). (C) Example

traces illustrating time course of active force measurements of EHD2 del/+ and del/del mesenteric arteries that were stimulated by phenylephrine (PE) and acetylcholine (ACh). Summary of data on relaxation (D) and contraction (E) of mesenteric arteries isolated from EHD2 del/+ and del/del mice (n(del/+) = 8/5; n(del/del) = 11/5). The data demonstrate reduced relaxation of EHD2 del/del arteries to ACh. Active force traces of EHD2 del/+ (F, H) or EHD2 del/del (G, H) mesenteric arteries treated with L-NAME (gray traces, untreated: n(del/+) = 17/5; n(del/del) = 36/5; L-NAME treated: n(del/+) = 8/5; n(del/del) = 5/5). (I) Application of NO donor SNP caused similar dose-dependent relaxations of EHD2 del/+ and del/del mesenteric arteries (n = 6, c(PE) = 1 μM). (J, K) Tail cuff blood pressure measurements of EHD2 del/+ and del/del mice (50 weeks old, n = 5) revealed no differences between EHD2 del/+ and del/del mice. Graphs illustrate each replicate with mean +/- SE, column bar and line graphs illustrate mean + SE, t-test or Mann Whitney U test were used to calculate significance, * P<0.05; n.s. not significant.

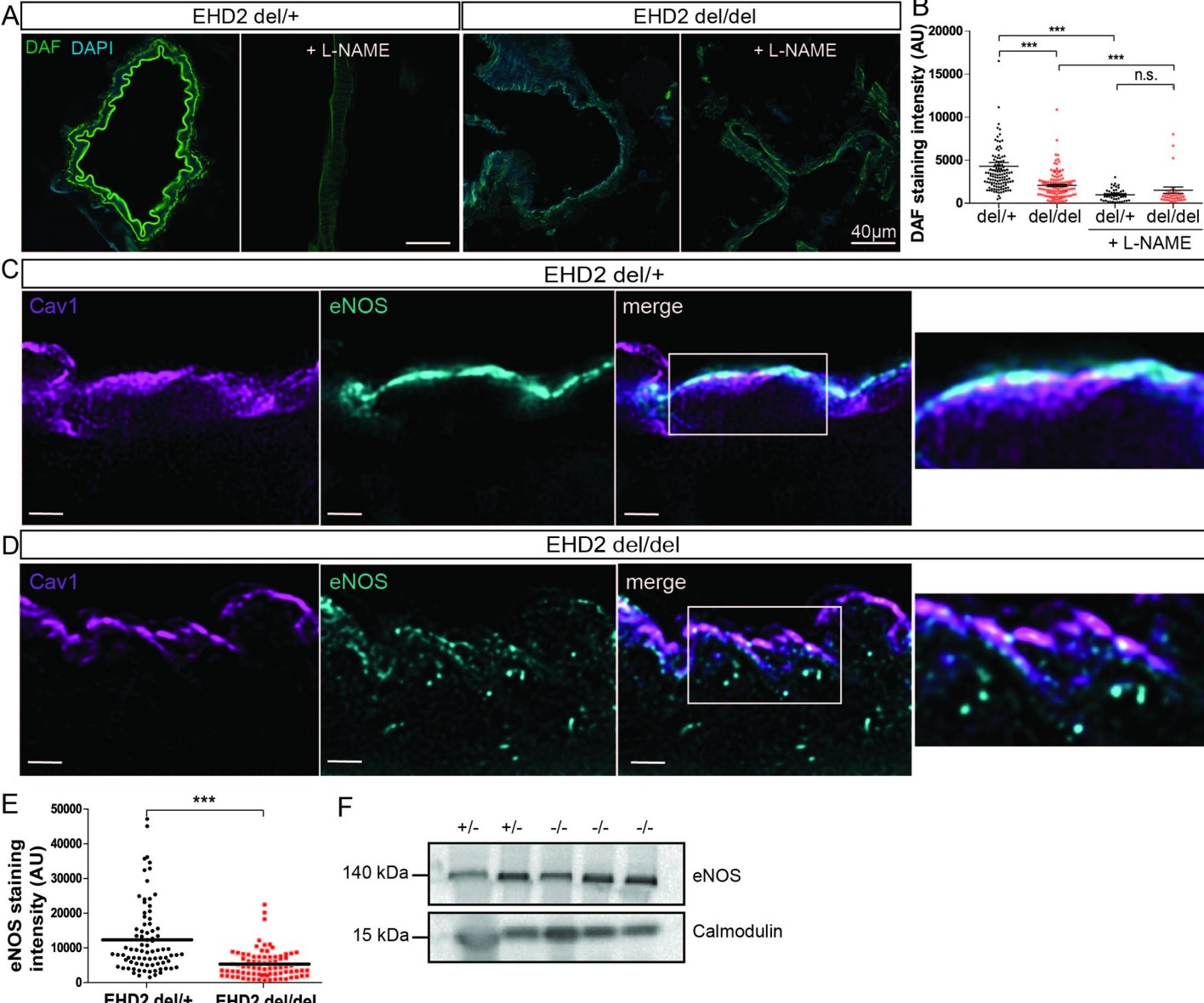

**Fig 3. Decreased NO concentrations and impairment of eNOS localization in EHD2 del/del mesenteric arteries.** (A) NO abundance was determined by DAF staining of EHD2 del/+ and del/del mesenteric arteries either untreated or pre-treated with L-NAME. (B) DAF staining intensity is significantly reduced for EHD2 del/del arteries compared to EHD2 del/+ (L-NAME: n(del/+) = 51/3; n(del/del) = 43/3; untreated: n(del/+) = 121/3; n(del/del) = 127/3). (C) STED imaging of eNOS (blue) and Cav1 (magenta) in EHD2 del/+ mesenteric artery cryostat sections illustrates eNOS localization at the plasma membrane of the small arteries in close proximity to caveolae (merge). (D) EHD2 del/del mesenteric arteries showed reduced eNOS staining at the plasma membrane, instead eNOS was detected within the cytosol of endothelial cells. Scale bar 4 μm. See also S4 Fig. (E) Plasma membrane eNOS staining was analyzed and revealed decreased eNOS fluorescence intensity in EHD2 del/del mesenteric arteries compared to EHD2 del/+ (n(del/+) = 83/3; n(del/del) = 81/3). (F) Analysis of total eNOS protein concentration in tissue lysates obtained from EHD2 del/+ and del/del small vessels by Western Blot. Calmodulin was used as loading control. Graphs illustrate each replicate with mean +/- SE, t-test or Mann Whitney U test were used to calculate significance, *** P<0.0001; n.s. not significant.

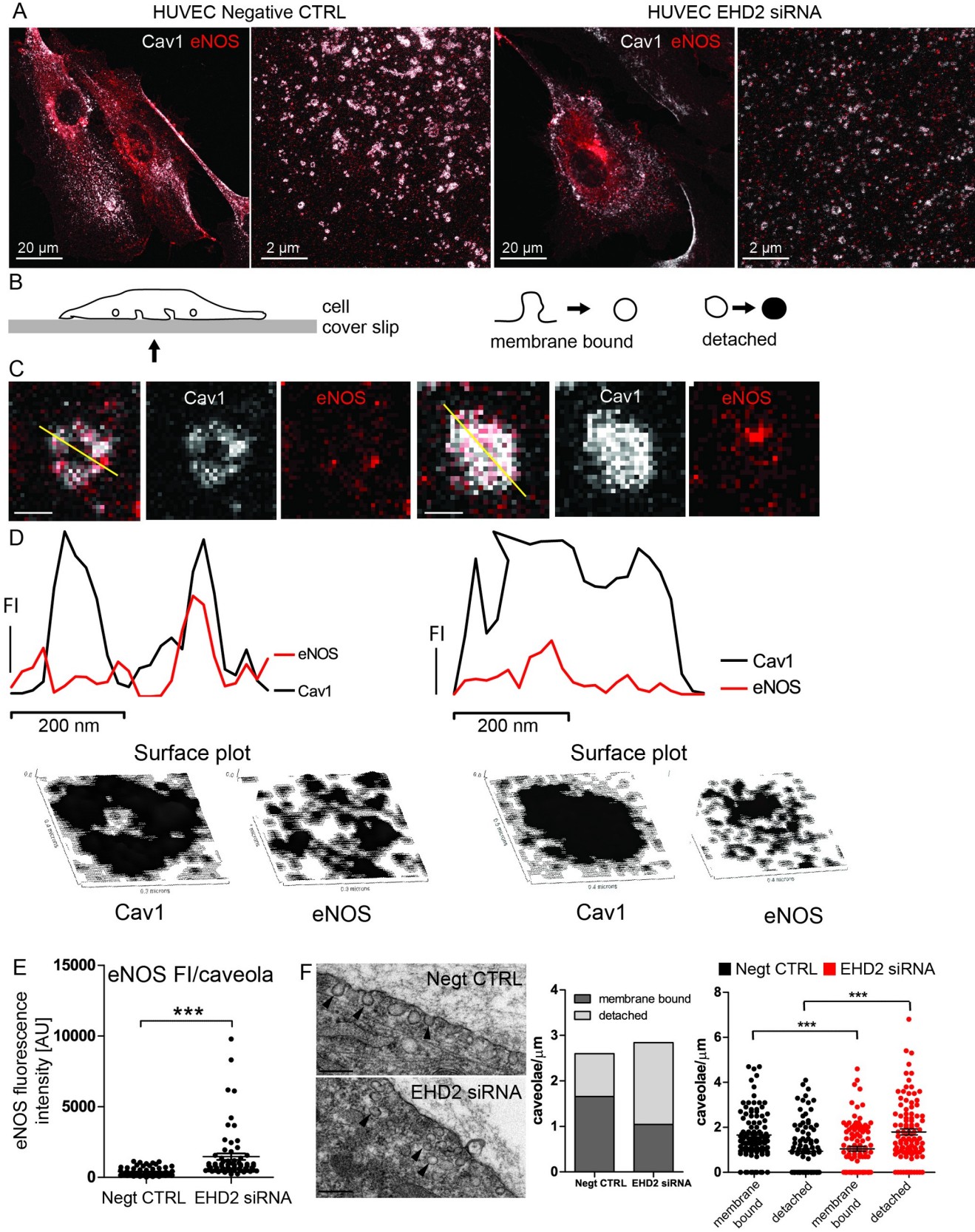

**Fig 4. EHD2 knockdown HUVECs showed eNOS localization in detached caveolae.** (A) HUVECs were stained against Cav1 (gray) and eNOS (red) and STED microscopy was used to analyze single caveolae. Overview of cells was obtained by confocal imaging (left image). STED was then applied on a 20 μm² area of the flat part of the cell (right image). (B, C) Membrane-bound caveolae appear in ring-like structures, detached caveolae from the plasma membrane are illustrated as filled vesicle-like shape (scale bar 200 nm). (D) Characteristic Cav1 fluorescence intensity (FI) plot profile of ring-like membrane bound and vesicle-like detached caveolae corresponds to surface plots and indicates for both caveolae shapes. (E) eNOS fluorescence intensity was measured for single caveolae in EHD2 control and EHD2 siRNA-treated HUVECs and revealed increased eNOS intensity in the absence of EHD2 (n(Negt. CTRL) = 61/2, n (EHD2 siRNA) = 64/2). (F) EM images illustrate characteristic caveolae shapes in HUVEC control and EHD2 siRNA treated cells, whereby EHD2 knockdown HUVECs revealed an increased number of detached caveolae (n(Negt CTRL) = 106/2, n(EHD2 si) = 104/2, scale bar 200 nm, graphs represent all replicates with mean+/- SE, Mann Whitney U test were used to calculate significance, *** P<0.0001).

In addition, we observed an increased total eNOS staining intensity in single caveolae of EHD2 knockdown cells compared to HUVEC control cells (Fig 4E). This experiment indicate that the loss of EHD2 results in increased fraction of Cav1-membrane bound eNOS suggesting a reduced eNOS activity. EM analysis further confirmed that EHD2 knockdown resulted in increased number of detached caveolae (Fig 4F).

## EHD2 knockdown in HUVECs decreased cellular NO concentration and cytosolic calcium

Similar to the experiments in small arteries, we also analyzed NO concentration in EHD2 knockdown HUVECs by DAF staining. Control cells revealed strong DAF staining intensity after ACh treatment in line with high cytosolic NO concentrations (Fig 5A). Pre-treatment with L-NAME completely abolished DAF staining. Consistent with our *in vivo* results, EHD2 knockdown in HUVECs resulted in decreased NO concentrations (Fig 5A and 5B).

It was previously reported that intracellular $Ca^{2+}$ release (from internal or extracellular stores) induces eNOS activity and, consequently, NO production [51]. To analyze the role of EHD2 dependent caveolae behavior on calcium signaling, cytosolic $Ca^{2+}$ was monitored in HUVECs treated with EHD2 siRNA (Fig 5C–5F). Control HUVECs revealed the characteristic 2-phase intracellular $Ca^{2+}$ increase upon stimulation with ATP (Fig 5C), very similar to published ATP or ACh-induced $Ca^{2+}$ events [52,53]. The first $Ca^{2+}$ peak has been assigned to intracellular $Ca^{2+}$ release from the ER, followed by the longer $Ca^{2+}$ plateau induced by extracellular $Ca^{2+}$ entry via $Ca^{2+}$ channels (store-operated $Ca^{2+}$ entry, SOCE). In contrast, EHD2 knockdown resulted in reduced $Ca^{2+}$ responses triggered by ATP treatment (Fig 5D). Compared to control HUVECs, the $Ca^{2+}$ events in HUVECs treated with EHD2 siRNA showed a reduced duration time (Fig 5C and 5E). The total $Ca^{2+}$ levels in HUVECs, as measured by Fura2 fluorescence intensity (excitation 380 nm), were not reduced upon EHD2 knockdown (Fig 5F). This indicates that intracellular $Ca^{2+}$ entry upon stimulation, likely via store operated calcium entry pathways, is affected in the absence of EHD2 [52].

As it was previously described that phosphorylation of Ser1177 increased eNOS activity [41,54], we investigated eNOS phosphorylation levels by Western blotting. Indeed, HUVECs treated with EHD2 siRNA showed reduced Ser1177 phosphorylation compared to control cells (Fig 5G). AKT phosphorylation was not impaired in EHD2 knockdown HUVEC compared to siRNA-treated HUVECs (Fig 5H). In summary, these data suggests that reduced $Ca^{2+}$ signaling and reduced phosphorylation of eNOS-Ser1177 lead to decreased eNOS activity in EHD2 knockdown cells (see also schematic overview in Fig 5I and S6 Fig).

## Reduced running wheel activity in EHD2 del/del mice

Although the loss of EHD2 *in vivo* resulted in severe impairment of vessel relaxation, the resting blood pressure in EHD2 del/del mice was not changed, as measured by tail-cuff measurements (Fig 2J and 2K). We therefore tested the physical constitution and behavior of EHD2

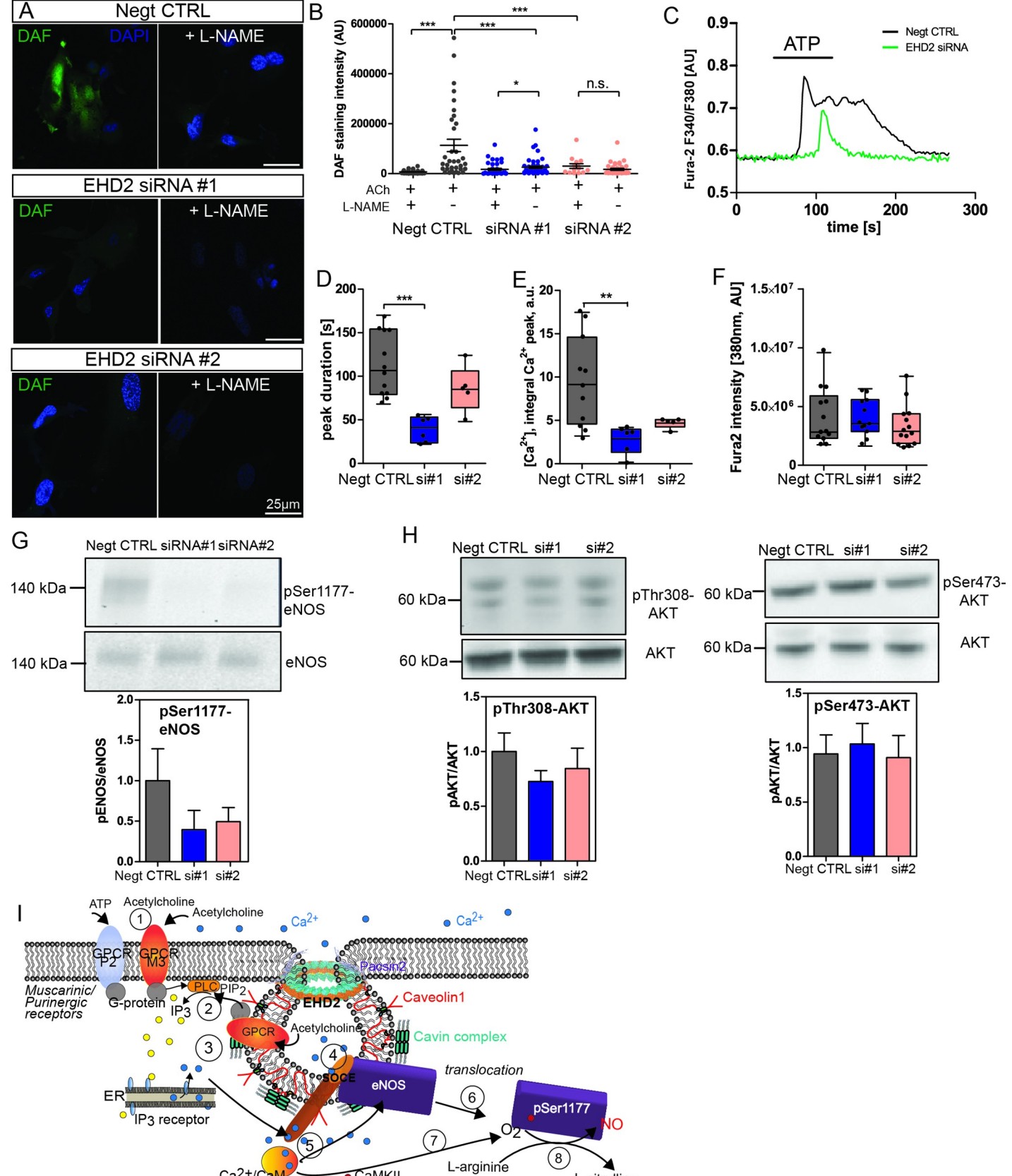

**Fig 5. EHD2 siRNA treated HUVEC showed decreased NO generation and cytosolic Ca²⁺ response.** (A) DAF staining of either non-sense siRNA or EHD2 siRNA (siRNA#1 or siRNA#2) treated HUVECs after acetylcholine stimulation revealed significantly reduced DAF fluorescence intensity in EHD2 siRNA-treated HUVECs. (B) Pre-incubation with L-NAME abolished NO production in all investigated HUVECs (L-NAME: n(Negt. CTRL) = 34/3, n(siRNA#1) = 37/3; n(siRNA#2) = 14/3;

untreated: n(Negt. CTRL) = 34/3, n(siRNA#1) = 46/3; n(siRNA#2) = 37/3)). (C) Cytosolic $Ca^{2+}$ recording in HUVEC treated with EHD2 siRNA. $Ca^{2+}$ response was triggered by 30 µM ATP as illustrated in the examples traces. (D, E) Reduced $Ca^{2+}$ peaks (as obtained by integration) and duration were observed in EHD2 knockdown HUVECs. (F) Fura2 fluorescence intensity excited at 380 nm was used as marker for overall cytosolic $Ca^{2+}$ load (n(Negt. CTRL) = 11/3, n(siRNA#1) = 6/3; n(siRNA#2) = 6/3). (G) Reduction of phosphorylation level of eNOS-Ser1177 in HUVEC treated with EHD2 siRNA compared to non-sense siRNA after acetylcholine stimulation (n(Negt. CTRL) = 8, n(siRNA#1) = 8; n(siRNA#2) = 7). (H) Phosphorylation of AKT-Thr308 and AKT-Ser473 in HUVEC revealed no difference between EHD2 siRNA or control siRNA treated cells (n(Negt. CTRL) = 10, n(siRNA#1) = 11; n(siRNA#2) = 10). (I) Schematic overview of acetylcholine-triggered $Ca^{2+}$ response and eNOS activation. After acetylcholine binding to the G protein coupled muscarinic M3 receptor or ATP binding to P2 purinergic receptor (1), PLC is activated and triggers $IP_3$ production (2). $IP_3$ binds to the $IP_3$ receptor within the ER membrane and induces local $Ca^{2+}$ release via $IP_3$ receptors (3), which correlates to the first peak observed in the $Ca^{2+}$ recordings. Local intracellular $Ca^{2+}$ close to the plasma membrane then activates $Ca^{2+}$ channels localized within caveolae (4, store operated $Ca^{2+}$ entry, SOCE). Increased $Ca^{2+}$ influx increases the cytosolic $Ca^{2+}$ concentration (second peak, longer duration). After $Ca^{2+}$ binds to calmodulin (CaM) (5), CaM binds to eNOS (6) followed by the disruption of eNOS-Cav1 interaction and translocation of eNOS into the cytosol. $Ca^{2+}$/CaM further induces autophosphorylation of CamKII resulting in phosphorylation of Ser1177 of eNOS (7). Activated eNOS catalyzes the conversion of L-arginine to L-citrulline leading to NO production (8). For comparison to the situation in EHD2 knockout cells, see S6 Fig. Graph illustrates each replicate with mean +/- SE, box plots indicate mean with whiskers from min to max, t-test or Mann Whitney U test were used to calculate significance, * P<0.05; ** P<0.001; *** P<0.0001; n.s. not significant.

del/del mice. Heart sections obtained from EHD2 del/del hearts did not reveal gross differences in tissue appearance (Fig 6A). Notably, echocardiographic recordings monitored a slightly increased left ventricular posterior wall thickness (LVPW, sys, 1.1 ± 0.1 mm for del/del vs. 0.9 ± 0.1 mm for del/+) and interventricular septum size (IVS, sys, 1.1 ± 0.1 mm for del/del vs. 0.9 ± 0.1 mm for del/+, Fig 6B) in 20 weeks old EHD2 del/del mice compared to del/+ mice (Fig 6B). Table 1 summarizes the specific echocardiographic parameters for EHD2 del/+ and del/del mice during the development.

The echocardiographic changes prompted us to evaluate the physical fitness of EHD del/del mice by exposing them to a voluntary running wheel exercise. In these experiments, EHD2 del/del mice (Fig 6C, red line) showed a reduced running activity compared to EHD2 del/+ mice that were running during the complete dark phase (Fig 6C, black line). Total running distance over 2 weeks was significantly reduced from 63 ± 4 km in EHD2 del/+ mice to 32 ± 6 km for EHD2 del/del mice (Fig 6D). These data suggest that EHD2 del/del mice have a reduced physical fitness required for sustained running activity. Interestingly, no difference was found in heart rate (Fig 6E), body weight (Fig 6F) or left ventricle wall size (Fig 6G, summary S1 Table) between EHD2 del/+ and del/del mice after exercise.

## Discussion

Cav1, the major caveolae protein, is an established regulator for eNOS activity in endothelial cells, and the molecular interaction of the two proteins has been studied in detail [14,39,55,56]. However, the physiological relevance of caveolae plasma membrane localization and stabilization in the context of vascular eNOS function has not been investigated. Here, we report that EHD2-dependent caveolae stabilization is required for correct function and regulation of eNOS. Upon loss of EHD2 in small vessels such as mesenteric arteries, the number of detached caveolae in the endothelium was increased. In addition, relaxation in these vessels was impaired due to insufficient NO production by eNOS. By super-resolution microscopy, we demonstrated that eNOS localization at the plasma membrane depends on the stable integration of caveolae. We also show that detached caveolae still contain eNOS.

So far, eNOS caveolae localization has been mainly studied by biochemical membrane isolation and fractionation [37,38,57] experiments and immunogold labeling [58,59]. In this study, we analyzed eNOS localization in small vessels by STED microscopy which allowed us to evaluate the spatial parameters in a more quantitative fashion and at higher resolution. Indeed, we observed eNOS localized at the plasma membrane in the endothelium in the vessel sections. In EHD2 del/+ mesenteric arteries, eNOS localized to the endothelial plasma membrane in close proximity to Cav1. Surprisingly, eNOS was found non-homogenously distributed in clusters at

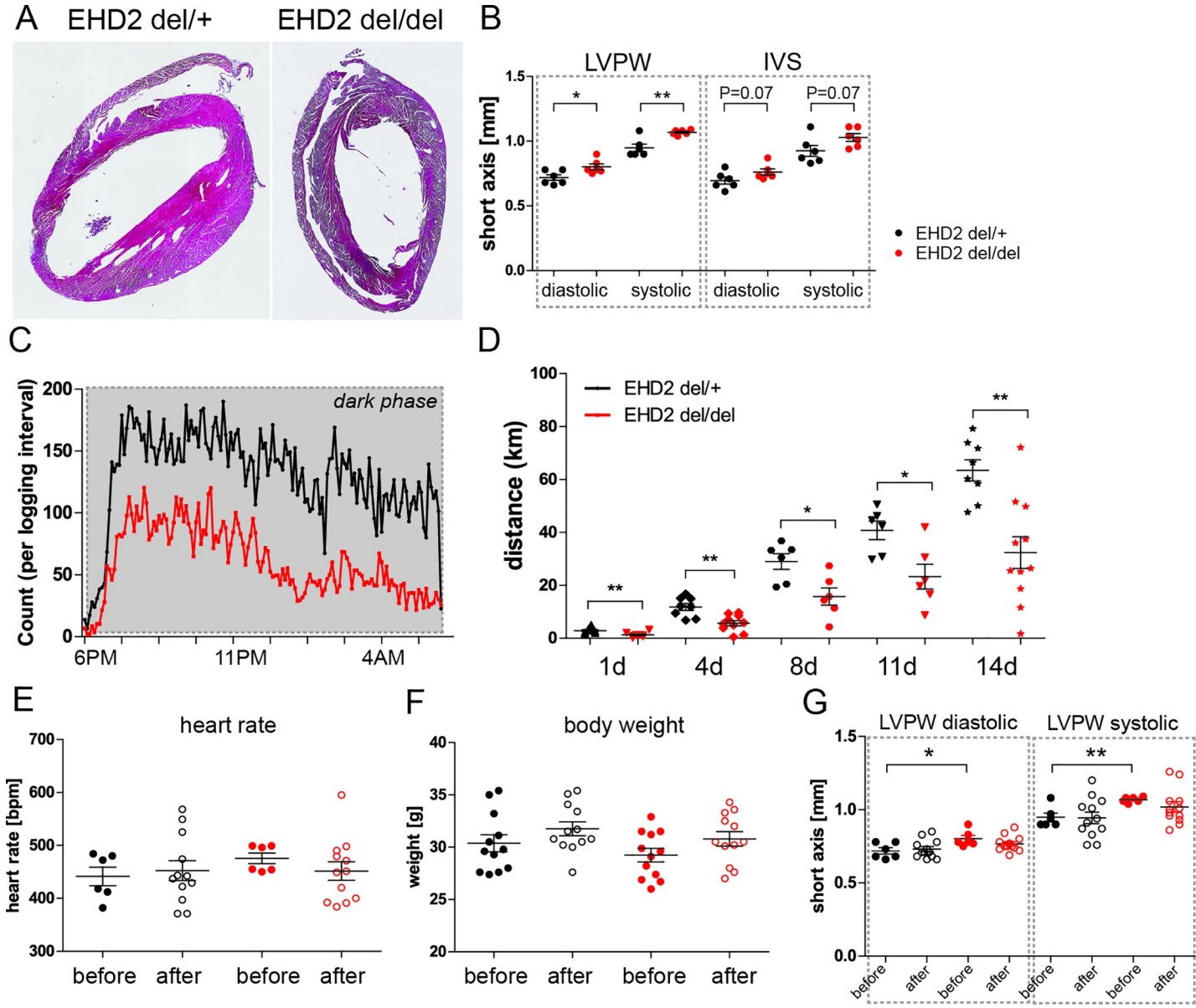

**Fig 6. EHD2 del/del mice showed reduced running wheel activity.** (A) Heart paraffin sections illustrated no apparent morphological changes in EHD2 del/del compared to EHD2 del/+ mice. (B) Echocardiography measurement of 20 weeks old mice showed an increased left ventricle wall size in EHD2 del/del mice (n = 6). (C) Recorded running wheel activity of EHD2 del/+ and del/del mice during the night phase (n(del/+) = 7, n(del/del) = 8). (D) Measured running distance of EHD2 del/+ and del/del mice over two weeks revealed significantly reduced total running distance for EHD2 del/del mice compared to the control group. (E) Heart rate of EHD2 del/+ and del/del mice before and after the running wheel training (n = 6–12) was not impaired. (F) Body weight was not influenced by the running wheel exercise in EHD2 del/+ and del/del mice (n = 6–12). (G) Left ventricle wall size is aligned after two weeks of running wheel training in EHD2 del/+ and del/del mice (n = 6–12). Graphs illustrate each replicate with mean +/- SE, t-test or Mann Whitney U test were used to calculate significance, * P<0.05; ** P<0.001. LVPW–left ventricular posterior wall thickness, IVS–interventricular septum.

the plasma membrane in control mice. In contrast, loss of EHD2 and therefore increased caveolae detachment from the membrane resulted in the redistribution of eNOS into the cytoplasm (Fig 3D and S4 Fig, see also summary S6 Fig). Notably, EHD2 knockdown HUVECs showed that detached caveolae still contained eNOS. This suggests that eNOS still interacts with Cav1 in internalized caveolae leading to eNOS inhibition, in agreement with the reduced NO levels observed in EHD2 siRNA-treated cells or EHD2 lacking arteries (Figs 3 and 5A).

**Table 1. Echocardiographic measurements of EHD2 del/+ and del/del mice.**

| | 10 weeks | | 20 weeks | | 1 year | |
|---|---|---|---|---|---|---|
| | EHD2 del/+ | EHD2 del/del | EHD2 del/+ | EHD2 del/del | EHD2 del/+ | EHD2 del/del |
| LV posterior wall thickness, dia [mm] | 0.7±0.1 | 0.7±0.1 | 0.7±0.1 | 0.8±0.1 | 0.9±0.1 | 0.8±0.1 |
| LV posterior wall thickness, sys [mm] | 1.0±0.2 | 1.0±0.1 | 0.9±0.1 | 1.1±0.1 | 1.0±0.1 | 1.1±0.1 |
| LV mass, uncorrected [mg] | 90±10 | 96±6 | 134±5 | 150±10 | 120±10 | 150 ±20 |
| Fraction shortening [%] | 30±6 | 28±1 | 22±2 | 21±2 | 23±2 | 23±3 |
| Ejection fraction [%] | 60±8 | 55±2 | 43±3 | 44±3 | 47±2 | 46±5 |
| Stroke volume [µl] | 30±2 | 32±2 | 32±2 | 30±2 | 33±3 | 36±4 |
| Cardiac output [ml/min] | 14±1 | 14.3±0.6 | 14±2 | 14±1 | 14±1 | 18±2 |
| Heart rate [bpm] | 460±10 | 460±7 | 440±20 | 480±10 | 430±20 | 460±20 |

This observation highlights the importance of caveolae plasma membrane stabilization, and not Cav1 expression alone, for correct eNOS localization and regulation. Sowa et al. (2001) already reported that Cav1 expression without formation of caveolae did not result in correct eNOS-Cav1 regulation [57]. Furthermore, in the complete absence of caveolae due to Cav1 deletion, increased eNOS activity was observed [60]. These results indicate that increased eNOS internalization due to caveolae internalization in EHD2 del/del cells mechanistically differs from eNOS translocation to the cytosol upon a physiological stimulus (see schematic summary in S6 Fig). We therefore conclude that caveolae stabilization is essential for correct eNOS localization at the plasma membrane and activation.

We found that EHD2 deletion caused impaired eNOS activity that consequently resulted in reduced relaxation of small vessels. Previous observations demonstrated that prostaglandin i2 (PGi2) production and release in blood vessels can also induce relaxation of arteries. However, compared to NO, PGi2 plays only a minor role during vasodilation [46]. Furthermore, increased PGi2 release would also result in increased relaxation of arteries which was not observed in EHD2 del/del mice. When analyzing the involvement of eNOS activation pathways, we observed a reduced phosphorylation of eNOS Ser1177 in EHD2 knockdown endothelial cells which is indicative of a reduced eNOS activity. Analysis of AKT phosphorylation levels (Ser473, Thr308) did not reveal major differences in HUVECs upon EHD2 knockdown compared to control cells (Fig 5G). However, activation of eNOS can be also triggered by increased cytosolic $Ca^{2+}$ concentration [35,51]. Surprisingly, EHD2 siRNA-treated HUVECs showed a strong reduction in ATP-triggered $Ca^{2+}$ responses (Fig 5C). Previously reported SOCE pathways depend on correct caveolae formation (Fig 5I) [15,18,19,52,61]. The reduced stimulated $Ca^{2+}$ responses in endothelial cells lacking EHD2 are therefore likely caused by an impaired SOCE response (S6 Fig). These results indicate that caveolae serve as a unique plasma membrane domain necessary for correct localization of various $Ca^{2+}$ channels. Consequently, destabilization and increased internalization of caveolae due to EHD2 loss leads to reduced cytosolic $Ca^{2+}$ responses.

Resting blood pressure, as measured by tail-cuff measurements, and heart function were not altered in EHD2 del/del mice. It should be noted that tail-cuff measurements are not the most accurate method for determining small changes in blood pressure. However, as we did not obtain any indication that the blood pressure is altered in EHD2 del/del compared to control mice, we discarded more sophisticated and invasive blood pressure measurements. Indeed, previously reported blood measurements in Cav1 knockout mice resulting in complete loss of caveolae also leads to only a moderate effect on blood pressure [16]. Apparently, under laboratory conditions, endothelial cells can largely cope with caveolae loss. Nevertheless, the observed reduced relaxation of EHD2 del/del vessels in response to acetylcholine indicate that

EHD2 deficiency might have effects on the physical appearance. Consistent with this idea, we detected a reduced running wheel activity of EHD2 del/del mice (Fig 6). We speculate that besides the impaired vessel function also increased lipid accumulation in EHD2 del/del mice [33] may contribute to fitness. Future studies with endothelial- or fat-specific EHD2 knockout mouse models will clarify the involved mechanism.

## Conclusion and outlook

We show in this manuscript that EHD2 at the caveolae neck is required for correct eNOS localization and signaling and therefore for proper endothelial function. In addition, we demonstrated that caveolae detachment from the plasma membrane results in decreased activation of eNOS via caveolae dependent $Ca^{2+}$ entry pathways. Reduced eNOS activity and NO production are hallmarks of endothelial dysfunction, as found, e.g., in type 2 diabetes and obesity [62–64]. We therefore envisage that altered EHD2 function may also be found in some of these diseases.

## Supporting information

**S1 Table. Echocardiography of 20 weeks old EHD2 del/+ and del/del mice before and after 2 weeks running wheel training.**
(DOCX)

**S1 Fig. EHD2 antibody staining in EHD2 del/+ vs. EHD2 del/del arteries.**
(TIF)

**S2 Fig. Angiogenesis in EHD2 del/+ and del/del mesenteric artery pieces.**
(TIF)

**S3 Fig. Functional analysis of EHD2 del/+ and del/del mesenteric arteries (related to Fig 2).**
(TIF)

**S4 Fig. STED imaging of eNOS and Cav1 in EHD2 del/+ and del/del mesenteric arteries (related to Fig 3).**
(TIF)

**S5 Fig. EHD2 siRNA knockdown in HUVEC (related to Figs 4 and 5).**
(TIF)

**S6 Fig. Model of EHD2 function in endothelial cells.**
(TIF)

**S7 Fig. Raw images of Western Blots files of eNOS and AKT phosphorylation in HUVECs.**
(TIF)

**S8 Fig. Raw images of Western Blots files of eNOS level in mesenteric arteries.**
(TIF)

**S9 Fig. Raw images of Western Blots files of HUVECs treated with EHD2 siRNA.**
(TIF)

## Acknowledgments

We thank Petra Stallerow for taking care of the EHD2 mouse strain, the Advanced Light Microscopy facility from MDC for their technical support, the Pathobiology platform from

Arndt Heuser from MDC for tail cuff measurements and electrocardiography, Frederike Kemplin for helping with the running wheel experiment, Marcus Semtner for the calcium imaging set up, Maria Kamprath for helping with histological heart sections and Holger Gerhardt for critically reading the manuscript.

## Author Contributions

**Conceptualization:** Claudia Matthaeus, Dominik N. Müller, Maik Gollasch, Oliver Daumke.

**Data curation:** Claudia Matthaeus, Séverine Kunz, Martin Lehmann, Cheng Zhong, Ines Lahmann.

**Formal analysis:** Cheng Zhong.

**Funding acquisition:** Maik Gollasch, Oliver Daumke.

**Investigation:** Claudia Matthaeus, Xiaoming Lian, Séverine Kunz, Martin Lehmann, Carola Bernert, Ines Lahmann.

**Methodology:** Claudia Matthaeus, Xiaoming Lian, Martin Lehmann, Ines Lahmann.

**Project administration:** Claudia Matthaeus.

**Supervision:** Maik Gollasch, Oliver Daumke.

**Validation:** Claudia Matthaeus, Martin Lehmann.

**Visualization:** Claudia Matthaeus.

**Writing – original draft:** Claudia Matthaeus, Oliver Daumke.

**Writing – review & editing:** Xiaoming Lian, Séverine Kunz, Martin Lehmann, Ines Lahmann, Dominik N. Müller, Maik Gollasch, Oliver Daumke.

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
