## [Decision Letter · Decision Letter 0]

18 Jul 2019

PONE-D-19-17247

MAX-DELBRÜCK-CENTRUM FÜR MOLEKULARE MEDIZIN BERLIN-BUCH

PLOS ONE

Dear Dr. Matthaeus,

Thank you for submitting your manuscript to PLOS ONE. After careful consideration, we feel that it has merit but does not fully meet PLOS ONE’s publication criteria as it currently stands. Therefore, we invite you to submit a revised version of the manuscript that addresses the points raised by the reviewers.

While addressing the reviewers' concerns please specifically pay attention to the following points: 1) please make sure that all your conclusions are supported by experimental data. For example, the statement about eNOS and Cav1 interaction in internal pools would require additional experimental confirmation. 2) It is unclear why calcium imaging data are provided for ATP in HUVEC, whereas acetylcholine was used in ex vivo experiments. 3) To comply with the new PLOS One policy, please provide the entire, un-cropped Western blot images in Supplemental figures for ALL Western blot data included in the manuscript. 4) For isometric tension experiments, please provide traces with the complete recovery after KCl-induced contractions. Why was not a nitric oxide donor used to confirm that NO could still dilate the pre-contracted EHD2 del/del vessels? 5) Please confirm that the pre-contracted vessels cannot be dilated with an alternative endothelial-dependent vasorelaxant and provide experimental evidence that impaired acetylcholine relaxation in EHD2 del/del is not due to a reduced plasma membrane protein expression of muscarinic receptors.               

We would appreciate receiving your revised manuscript by Sep 01 2019 11:59PM. To enhance the reproducibility of your results, we recommend that if applicable you deposit your laboratory protocols in protocols.io, where a protocol can be assigned its own identifier (DOI) such that it can be cited independently in the future. For instructions see: http://journals.plos.org/plosone/s/submission-guidelines#loc-laboratory-protocols

A rebuttal letter that responds to each point raised by the academic editor and reviewers. This letter should be uploaded as separate file and labeled 'Response to Reviewers'.A marked-up copy of your manuscript that highlights changes made to the original version. This file should be uploaded as separate file and labeled 'Revised Manuscript with Track Changes'.An unmarked version of your revised paper without tracked changes. This file should be uploaded as separate file and labeled 'Manuscript'.

We look forward to receiving your revised manuscript.

Kind regards,

Alexander G. Obukhov, Ph.D.

Academic Editor

PLOS ONE

2. Please amend the title either on the online submission form or in your manuscript so that they are identical.

Reviewers' comments:

Reviewer's Responses to Questions

**Comments to the Author**

1. Is the manuscript technically sound, and do the data support the conclusions?

Reviewer #1: Partly

Reviewer #2: Partly

Reviewer #3: Yes

2. Has the statistical analysis been performed appropriately and rigorously? 

Reviewer #1: Yes

Reviewer #2: Yes

Reviewer #3: Yes

3. Have the authors made all data underlying the findings in their manuscript fully available?

Reviewer #1: Yes

Reviewer #2: Yes

Reviewer #3: Yes

4. Is the manuscript presented in an intelligible fashion and written in standard English?

Reviewer #1: Yes

Reviewer #2: Yes

Reviewer #3: Yes

5. Review Comments to the Author

Reviewer #1: This study describes a connection between EHD2, caveolae and NO signaling. The study uses mostly immunofluorescence to look at colocalization along with other methods to show the importance of formed caveolae structures on NO signaling. The data and results are very nice with two exceptions that need to be resolved:

1 – The STED images are confusing. The images do not resemble the structures seen by high resolution methods.

2 - The cartoon in Figure 4 is not accurate. The G proteins that are responsible for calcium signals localize in caveolae which enhances calcium signals. The authors need to revise their model.

Reviewer #2: This article makes numerous interesting observations that link the physiological function of eNOS with the structural retention of caveolae at the PM. The authors demonstrate that loss of EHD2 results in functional changes in eNOS, impaired mesenteric artery relaxation, reduced NO synthesis, and changes in Ca2+ signalling in HUVEC cells that they claim are a consequence of re-distribution of caveolae away from the cell surface. The authors go on to demonstrate that loss of EHD2 does not affect viability of mice nor impact upon blood pressure but does affect the exercise fitness of the mice. The data and model put forward are of importance to the field and would be of interest to the readers of PLOS ONE. But there are a few key points to address which make the manuscript unsuitable for publication is its current state.

Major Points:

The authors claim that the loss of EHD2 from endothelial tissue results in a reduction in the total eNOS at the cell surface by STED microscopy on cryostat tissue sections. These findings are critical for the manuscript and their proposed model but the images provided are not as clear as the authors suggest. The authors claim that Figure 3 and S3 show a redistribution of eNOS to an internal pool in EHD2 KO arteries and that this can be quantified by “eNOS staining intensity” at the plasma membrane (Fig 3E). It is unclear from their methods how was this analysis performed - were subregions selected and analysed for fluorescence intensity? If so, Fig S1 demonstrates “normal” surface association of eNOS in arteries in EHD2 KO in some cells and reduced association in others – critically this reduction in PM intensity of eNOS also observed in the control EHD2 -/+ mice (S1A). To make these claims biochemical analyses are required to determine if there is a quantitative reduction in eNOS membrane association.

Moreover, the authors highlight the redistribution of eNOS from the PM to internal pools. In the discussion the authors suggest eNOS and Cav1 likely still interact in internal pools (i.e. on dissociated caveolae observed by their electron microscopy). However, from the images provided it is unclear if Cav1 and eNOS do interact away from the surface as these puncta appear to lack Cav1. This is an important point and one that should be addressed using the HUVEC cell model and STED microscopy.

Does loss of EHD2 in HUVEC cells affect the PM association of caveolae by electron microscopy?

Minor Corrections:

The affiliation of the authors has come up as the title of the article. I’m unsure if this is an error in the portal or a mistake by the authors.

Line 362: The sentence reads “Perfused and fixated…” this should read “Perfused and fixed…”

Reviewer #3: This well-written manuscript investigates Dynamin-related ATPase EHD2 in caveolae formation and eNOS NO release and vasodilation in vivo, ex vivo, as well as in HUVEC in vitro. The authors confirm the expression of EHD2 in adipocytes and vasculature and HUVEC, showing co-localization with eNOS. Presence of caveolae was confirmed in WT and EHD2 KO by EM, with significant changes in caveolae localization. Basic KCL and ACh vasoreactivity assays show a major decrease in Ach-induced vasorelaxation, with no effect of l-Name, suggesting reduced NO-dependent vasorelaxation. DAF imaging for NO show decrease staining in absence of EHD2 in tissues, and lower NO release and cytosolic Ca+ concentration supported these conclusion in HUVEC treated with EHD2 siRNA.

Major:

The way the results are described in the Results and Figure legends section is simply to succinct. There is no %, fold changes, there is simply not enough detail. Description of controls are barely discussed and sometimes omitted. Results should be better described, Legends should be more complete.

A decrease in vasorelaxation to ACh could also suggest a decrease in ACh muscarinic receptor, and therefore requires confirmation with vasorelaxation with another endothelial-dependent agonist. But this was not performed.

How about PGi2 on the myograph? Were these experiment performed in presence of a PGI2 synthesis inhibitor?

Minor:

Intro: 'Currently, it is not known if caveolae membrane domains or Cav1 alone is essential for correct eNOS localization and regulation. ' I think this sentence is inaccurate and requires an update. Work on mutant Cav1 has demystified the caveolae vs caveolin signaling question with regards to eNOS. (Sessa CircRes 2016, JCI 2010).

6. PLOS authors have the option to publish the peer review history of their article (what does this mean?). If published, this will include your full peer review and any attached files.

Reviewer #1: No

Reviewer #2: No

Reviewer #3: No

---

## [Author Response · Author response to Decision Letter 0]

31 Aug 2019

We would like to thank the editor and the three reviewers for their careful reading and the constructive comments which have helped to significantly improve our manuscript. Based on their comments, we carried out new experiments (please see new Fig. 2I, Fig. 4 and S3) and modified the manuscript at several positions, as indicated in our detailed point-by-point response below. We hope that with these changes, the manuscript can now be accepted for publication in PLOS ONE.

While addressing the reviewers' concerns please specifically pay attention to the following points:

1) Please make sure that all your conclusions are supported by experimental data. For example, the statement about eNOS and Cav1 interaction in internal pools would require additional experimental confirmation. 

We included additional STED data showing eNOS localization in detached caveolae in EHD2 knockdown HUVECs (Fig. 4). Furthermore, we toned down our conclusions on eNOS localization in isolated EHD2 del/del mesenteric arteries.

2) It is unclear why calcium imaging data are provided for ATP in HUVEC, whereas acetylcholine was used in ex vivo experiments.

Both acetylcholine and ATP are standardly and exchangeable used in the literature to activate eNOS via related signaling pathways (da Silva et al., Circulation, 2009, 119:871-879; Lohman et al., Cardiovas Research, 2012; 95:269-280). To allow comparability of our results to the existing literature, we followed in our manuscript protocols that were established for ATP-induced Ca2+ imaging specifically in HUVECs (Gifford et al., Journal of Endocrinology, 2004, 182:485-499; Lim et al., Placenta, 2015, 36:759-66; Sue et al., Pfluegers Archive Eur J Phy, 1999; Isshiki et al., PNAS, 1998, 95:5009-5014). Indeed, control HUVEC experiments showed the typically Ca2+ responses after ATP treatment as previously published (Nilius & Droogmans, Phys rev, 2001, 81:1416-1459; Gifford et al., Journal of Endocrinology, 2004, 182:485-499).

For the wire myography/ active force measurements, acetylcholine has been standardly used in our research (Järve et al, Exp Neurol. 2019, Lian et al, Front Physiol 2018, Szijártó et al. Hypertension 2018, Schmid et al., J Am Heart Assoc, 2018, etc.). Again, we followed established protocols to allow comparability of the results to previous data. 

3) To comply with the new PLOS One policy, please provide the entire, un-cropped Western blot images in Supplemental figures for all Western blot data included in the manuscript. 

All Western Blots with markers are included in Fig S7-9.

4) For isometric tension experiments, please provide traces with the complete recovery after KCl-induced contractions. Why was not a nitric oxide donor used to confirm that NO could still dilate the pre-contracted EHD2 del/del vessels?

As requested, we now included additional NO donor experiment to confirm general eNOS/NO downstream signaling functionality in EHD2 del/del arteries (Fig. 2I). Also as requested, we have included traces with complete recovery after KCl-induced contractions (Fig. S3A). The traces in Fig. 2 did not show a total recovery which requires more than 30 min. As only freshly prepared mesenteric artery pieces can be used for this experiment, we reduced the recovery time for KCl induced contraction to increase the number of experiments. We were therefore able to test 7-10 mesenteric artery pieces from one mice per day. Furthermore, after the KCl stimulus, the arteries were not used anymore. This is now mentioned in the legend of Fig. S3.

5) Please confirm that the pre-contracted vessels cannot be dilated with an alternative endothelial-dependent vasorelaxant and provide experimental evidence that impaired acetylcholine relaxation in EHD2 del/del is not due to a reduced plasma membrane protein expression of muscarinic receptors.

Our additional experiments demonstrate that phenylephrine and KCl-induced contractions and SNP-induced relaxations were normal in EHD2 del/del arteries. We provide a large number of different experiments demonstrating that EHD2 deficiency causes impaired eNOS activity (reduced NO levels in EHD2 knockout arteries and knockdown HUVEC, reduced eNOS phosphorylation in EHD2 siRNA HUVEC), which consequently results in reduced relaxation of small vessels. We also performed additional experiments indicating that muscarinic acetylcholine M3 receptor protein level is not altered in EHD2 del/del arteries (Fig. S3B). Together, we believe that the data present firm evidence that EHD2-dependent caveolae plasma membrane stabilization regulates correct eNOS localization and function in small arteries. As a consequence, insufficient NO production by the eNOS pathway is the primary cause of impaired relaxation observed in EHD2 del/del arteries.

1.) Please ensure that your manuscript meets PLOS ONE's style requirements, including those for file naming.

We changed the manuscript and the figures accordingly to the journal’s requirements.

2.) Please amend the title either on the online submission form or in your manuscript so that they are identical.

The title in both forms is now ‘eNOS-NO-induced small blood vessel relaxation requires EHD2-dependent caveolae stabilization’.

Reviewers' comments:

Reviewer #1: 

This study describes a connection between EHD2, caveolae and NO signaling. The study uses mostly immunofluorescence to look at colocalization along with other methods to show the importance of formed caveolae structures on NO signaling. The data and results are very nice with two exceptions that need to be resolved: 

Thank you very much for your excitement, we are very grateful for your comments.

1 – The STED images are confusing. The images do not resemble the structures seen by high resolution methods.

Compared to electron microscopy, STED analysis, especially in tissue sections, lacks resolution resulting in more diffuse images. The included STED images of mesenteric arteries illustrate eNOS and Cav1 localization, but the images do not show specific plasma membrane shapes, as seen in our EM analysis. In addition, distinct caveolae membrane forms cannot be distinguished. However, we observed a mislocalization of eNOS from the plasma membrane into the cytosol in EHD2 lacking tissue which based on our subsequent analysis most likely corresponds to detached caveolae. This observation was not seen with EM because the eNOS antibody did not work in immunogold-EM (we tested this before). Thus, in tissue sections, STED imaging was used to illustrate the mislocalization of eNOS, but details of the plasma membrane and caveolar shape could not be resolved.

To further characterize eNOS localization in caveolae, we now included new STED microscopy analysis of EHD2 knockdown HUVECs (Fig. 4), which allows us to visualize caveolae at much higher resolution compared to tissue sections. In these experiments, we can now resolve single caveolae. We show that detached caveolae still contain eNOS, therefore further supporting our previous data and our model. 

2 - The cartoon in Figure 4 is not accurate. The G proteins that are responsible for calcium signals localize in caveolae which enhances calcium signals. The authors need to revise their model.

Thank you very much for this valid concern. We changed the figure accordingly and also depicted G proteins within caveolae (Fig. 5).

Reviewer #2: This article makes numerous interesting observations that link the physiological function of eNOS with the structural retention of caveolae at the PM. The authors demonstrate that loss of EHD2 results in functional changes in eNOS, impaired mesenteric artery relaxation, reduced NO synthesis, and changes in Ca2+ signaling in HUVEC cells that they claim are a consequence of re-distribution of caveolae away from the cell surface. The authors go on to demonstrate that loss of EHD2 does not affect viability of mice nor impact upon blood pressure but does affect the exercise fitness of the mice. The data and model put forward are of importance to the field and would be of interest to the readers of PLOS ONE. But there are a few key points to address which make the manuscript unsuitable for publication is its current state.

Thank you very much for the overall positive assessment. We sought to address your concerns, as detailed below.

Major Points:

The authors claim that the loss of EHD2 from endothelial tissue results in a reduction in the total eNOS at the cell surface by STED microscopy on cryostat tissue sections. These findings are critical for the manuscript and their proposed model but the images provided are not as clear as the authors suggest. The authors claim that Figure 3 and S3 show a redistribution of eNOS to an internal pool in EHD2 KO arteries and that this can be quantified by “eNOS staining intensity” at the plasma membrane (Fig 3E). It is unclear from their methods how was this analysis performed - were subregions selected and analyzed for fluorescence intensity? If so, Fig S1 demonstrates “normal” surface association of eNOS in arteries in EHD2 KO in some cells and reduced association in others – critically this reduction in PM intensity of eNOS also observed in the control EHD2 -/+ mice (S1A). To make these claims, biochemical analyses are required to determine if there is a quantitative reduction in eNOS membrane association.

Thank you very much for this valid point. We now include a detailed description of the analysis in the methods part (please see immunohistology of cryostat sections). Indeed, we analyzed the plasma membrane region of the small mesenteric arteries in the STED images to determine eNOS staining intensity (Fig. 3E). Using this method, we observed reduced eNOS staining at the plasma membrane of endothelial cells in EHD2 del/del mesenteric arteries. In addition, we found distinct eNOS spots in the cytosol in EHD2 del/del small arteries, suggesting that eNOS is re-localizing from the plasma membrane into the cytosol.

However, the initial inspection of blood vessels by confocal microscopy (Fig. S1B, D) did not reveal a difference of eNOS localization in EHD2 knockout arteries because the resolution limit is 200 nm. This is now noted in the figure legend S1. Therefore, plasma membrane and cytosol cannot be distinguished in these images (Fig S1B, D). We exchanged the example image for EHD2 del/+ aorta (Fig. S1B) to better illustrate the homogenous eNOS staining in the endothelial cell layer of the vessels. 

Interestingly, our STED analysis revealed clustered eNOS localization at the plasma membrane in EHD2 control mice (Fig. 3). Compared to EHD2 lacking mice, eNOS staining intensity was strongly increased but we also observed sub-regions of plasma membrane with reduced eNOS staining suggesting eNOS might not be completely homogenously distributed at the endothelial cell surface. Please see discussion about eNOS localization. 

Moreover, the authors highlight the redistribution of eNOS from the PM to internal pools. In the discussion the authors suggest eNOS and Cav1 likely still interact in internal pools (i.e. on dissociated caveolae observed by their electron microscopy). However, from the images provided it is unclear if Cav1 and eNOS do interact away from the surface as these puncta appear to lack Cav1. This is an important point and one that should be addressed using the HUVEC cell model and STED microscopy.

Indeed, the STED images of the mesenteric arteries (Fig. 3) do not clearly demonstrate that the relocated eNOS still interacts with Cav1. To address this concern, we now included higher resolution STED analysis of eNOS in relation to Cav1 in EHD2 knockdown HUVECs (Fig. 4). We can now distinguish between membrane-bound and detached caveolae (Fig. 4B, C). Furthermore, we observed eNOS staining in plasma membrane-bound but also in detached caveolae. Compared to control HUVECs, EHD2 siRNA treated cells showed an increased eNOS intensity in their analyzed caveolae (Fig. 4E) suggesting that eNOS still localizes in close proximity to Cav1. 

Does loss of EHD2 in HUVEC cells affect the PM association of caveolae by electron microscopy?

We performed new EM analysis to address this concern (see Fig. 4F). Indeed, EM inspection of EHD2 knockdown HUVECs also revealed an increased number of detached caveolae (Fig. 4F), illustrating again that loss of EHD2 results in altered caveolae behavior and re-localization from the plasma membrane, similar to what we observed in our STED analysis (Fig. 4).

Minor Corrections:

The affiliation of the authors has come up as the title of the article. I’m unsure if this is an error in the portal or a mistake by the authors.

We apologize for this and changed the title in the submission form.

Line 362: The sentence reads “Perfused and fixated…” this should read “Perfused and fixed…”

Thank you, we changed this accordingly.

Reviewer #3: This well-written manuscript investigates Dynamin-related ATPase EHD2 in caveolae formation and eNOS NO release and vasodilation in vivo, ex vivo, as well as in HUVEC in vitro. The authors confirm the expression of EHD2 in adipocytes and vasculature and HUVEC, showing co-localization with eNOS. Presence of caveolae was confirmed in WT and EHD2 KO by EM, with significant changes in caveolae localization. Basic KCL and ACh vasoreactivity assays show a major decrease in Ach-induced vasorelaxation, with no effect of l-Name, suggesting reduced NO-dependent vasorelaxation. DAF imaging for NO show decrease staining in absence of EHD2 in tissues, and lower NO release and cytosolic Ca+ concentration supported these conclusion in HUVEC treated with EHD2 siRNA.

Thank you very much for the positive review. Please see below for our responses. 

Major:

The way the results are described in the Results and Figure legends section is simply to succinct. There is no %, fold changes, there is simply not enough detail. Description of controls are barely discussed and sometimes omitted. Results should be better described, Legends should be more complete.

We included the requested description of all experiments and also figure legends were extended. Please see the revised manuscript (new text/description is added in red).

A decrease in vasorelaxation to ACh could also suggest a decrease in ACh muscarinic receptor, and therefore requires confirmation with vasorelaxation with another endothelial-dependent agonist. But this was not performed.

In response to this concern, we further included ACh muscarinic M3 receptor analysis by staining mesenteric artery cryosections isolated from EHD2 del/+ and del/del mice (Fig. S3B). Indeed, we did not observe an altered M3 staining for EHD2 lacking arteries indicating that the M3 protein level is not reduced in these vessels. 

Bradykinin is another agonist that is used to analyze vasorelaxation. However, our own experience and other previous studies showed that bradykinin induced relaxation is not stable (for example, see Wirth et al., 1996, 354:38-43). Therefore, we did not perform this experiment. Instead, we now included additional experiments using a NO donor (SNP) to evaluate whether EHD2 deficiency affects eNOS function vs. downstream signaling. The results are presented in Fig. 2I and discussed in the text. The new data demonstrate that downstream signaling of eNOS is intact in EHD2 del/del arteries. These data support the general concept that EHD2 deficiency causes impaired eNOS activity that consequently results in reduced relaxation of small vessels. Our additional experiments demonstrate that PE and KCl-induced contractions and SNP-induced relaxations were normal in EHD2 del/del arteries. We provide a large number of different experiments demonstrating that EHD2 deficiency causes impaired eNOS activity (reduced NO amount in EHD2 lacking tissue or cells, reduced eNOS phosphorylation), which consequently results in reduced relaxation of small vessels. Together, we believe that the data present firm evidence that EHD2-dependent caveolae plasma membrane stabilization regulates correct eNOS localization and function in small arteries. As a consequence, insufficient NO production by the eNOS pathway is the primary cause of impaired relaxation observed in EHD2 del/del arteries.

How about PGi2 on the myograph? Were these experiment performed in presence of a PGI2 synthesis inhibitor?

Previous experiments from us showed that prostaglycin (PGi2) is not produced during acetylcholine relaxation in mesenteric arteries (Hercule et al., Arteriosclerosis, thrombosis, and Vascular Biology 2009, 29:54-60). Therefore, we did not included COX inhibitors in the experimental set up. This is mentioned now in the methods.

It was suggested that PGi2 can be produced and released in arteries. However, in general it only plays a minor role during vasodilation compared to NO (Majed & Khalil, Pharmacol. Rev., 2012, 64:540-582). Nevertheless, increased PGi2 release would also result in increased relaxation of arteries which was not observed in EHD2 del/del mice in our study. In contrast, we observed a strong inhibition of mesenteric relaxation and therefore excluded this possibility. This is now discussed. 

Minor:

Intro: 'Currently, it is not known if caveolae membrane domains or Cav1 alone is essential for correct eNOS localization and regulation. ' I think this sentence is inaccurate and requires an update. Work on mutant Cav1 has demystified the caveolae vs caveolin signaling question with regards to eNOS. (Sessa CircRes 2016, JCI 2010).

We concur, indeed, the Sessa lab determined the relationship between Cav1 and eNOS in great detail. We therefore removed the sentence from the introduction.

---

## [Decision Letter · Decision Letter 1]

25 Sep 2019

eNOS-NO-induced small blood vessel relaxation requires EHD2-dependent caveolae stabilization

PONE-D-19-17247R1

Dear Dr. Matthaeus,

We are pleased to inform you that your manuscript has been judged scientifically suitable for publication and will be formally accepted for publication once it complies with all outstanding technical requirements.

With kind regards,

Alexander G. Obukhov, Ph.D.

Academic Editor

PLOS ONE

Reviewers' comments:

Reviewer's Responses to Questions

**Comments to the Author**

1. If the authors have adequately addressed your comments raised in a previous round of review and you feel that this manuscript is now acceptable for publication, you may indicate that here to bypass the “Comments to the Author” section, enter your conflict of interest statement in the “Confidential to Editor” section, and submit your "Accept" recommendation.

Reviewer #1: All comments have been addressed

Reviewer #2: All comments have been addressed

Reviewer #3: All comments have been addressed

2. Is the manuscript technically sound, and do the data support the conclusions?

Reviewer #1: Yes

Reviewer #2: Yes

Reviewer #3: Yes

3. Has the statistical analysis been performed appropriately and rigorously? 

Reviewer #1: Yes

Reviewer #2: Yes

Reviewer #3: Yes

4. Have the authors made all data underlying the findings in their manuscript fully available?

Reviewer #1: Yes

Reviewer #2: Yes

Reviewer #3: Yes

5. Is the manuscript presented in an intelligible fashion and written in standard English?

Reviewer #1: Yes

Reviewer #2: Yes

Reviewer #3: Yes

6. Review Comments to the Author

Reviewer #1: (No Response)

Reviewer #2: (No Response)

Reviewer #3: I am satisfied by the revised version of the manuscript, this paper is ready for publication.

7. PLOS authors have the option to publish the peer review history of their article (what does this mean?). If published, this will include your full peer review and any attached files.

Reviewer #1: No

Reviewer #2: No

Reviewer #3: No

---

## [Editor Report · Acceptance letter]

2 Oct 2019

PONE-D-19-17247R1 

eNOS-NO-induced small blood vessel relaxation requires EHD2-dependent caveolae stabilization 

Dear Dr. Matthaeus:

I am pleased to inform you that your manuscript has been deemed suitable for publication in PLOS ONE. Congratulations! Your manuscript is now with our production department. 

With kind regards,

on behalf of

Dr. Alexander G Obukhov 

Academic Editor

PLOS ONE